# The Potential of Low-Cost UAVs and Open-Source Photogrammetry Software for High-Resolution Monitoring of Alpine Glaciers: A Case Study from the Kanderfirn (Swiss Alps)

**Alexander R. Groos** [1,*], **Thalia J. Bertschinger** [1]**, Céline M. Kummer** [1]**, Sabrina Erlwein** [2]**, Lukas Munz** [1] **and Andreas Philipp** [3]

1   Institute of Geography, University of Bern, Hallerstrasse 12, 3012 Bern, Switzerland
2   Chair for Strategic Landscape Planning and Management, Technical University of Munich, Emil-Ramann-Str. 6, 85354 Freising, Germany
3   Institute of Geography, University of Augsburg, Universitätsstr. 10, 86135 Augsburg, Germany
*   Correspondence: alexander.groos@giub.unibe.ch; Tel.: +41-31631-8019

**Abstract:** Unmanned Aerial Vehicles (UAV) are a rapidly evolving tool in geosciences and are increasingly deployed for studying the dynamic processes of the earth's surface. To assess the potential of autonomous low-cost UAVs for the mapping and monitoring of alpine glaciers, we conducted multiple aerial surveys on the Kanderfirn in the Swiss Alps in 2017 and 2018 using open hardware and software of the Paparazzi UAV project. The open-source photogrammetry software OpenDroneMap was tested for the generation of high-resolution orthophotos and digital surface models (DSMs) from aerial imagery and cross-checked with the well-established proprietary software Pix4D. Accurately measured ground control points served for the determination of the geometric accuracy of the orthophotos and DSMs. A horizontal (xy) accuracy of 0.7–1.2 m and a vertical (z) accuracy of 0.7–2.1 m was achieved for OpenDroneMap, compared to a xy-accuracy of 0.3–0.5 m and a z-accuracy of 0.4–0.5 m obtained for Pix4D. Based on the analysis and comparison of different orthophotos and DSMs, surface elevation, roughness and brightness changes from 3 June to 29 September 2018 were quantified. While the brightness of the glacier surface decreased linearly over the ablation season, the surface roughness increased. The mean DSM-based elevation change across the glacier tongue was 8 m, overestimating the measured melting and surface lowering at the installed ablation stakes by about 1.5 m. The presented results highlight that self-built fixed-wing UAVs in tandem with open-source photogrammetry software are an affordable alternative to commercial remote-sensing platforms and proprietary software. The applied low-cost approach also provides great potential for other regions and geoscientific disciplines.

**Keywords:** glacier monitoring; glacier dynamics; Unmanned Aerial Vehicles; Paparazzi UAV; OpenDroneMap; photogrammetry; structure from motion; orthophotos; digital elevation models

## 1. Introduction

Global climate change affects glaciers worldwide and has led to an increased ice mass loss in recent decades (e.g., [1,2]). Ice mass loss contributes to sea-level rise (e.g., [3–5]) and alters the hydrological cycle as well as seasonal fresh-water availability in many regions around the globe (e.g., [6–8]). For quantification of the ongoing ice mass loss and projection of future runoff regimes, monitoring and analysing spatiotemporal changes in the geometry, mass budget, dynamics and surface characteristics (albedo, roughness, debris thickness, etc.) of glaciers is fundamental.

Before the era of satellite observations, glacier monitoring primarily relied on in-situ mass balance measurements at a few sites (e.g., [9,10]). To complement point measurements and study larger areas of the cryosphere, the application of satellite-based and airborne remote sensing data has now become standard (e.g., [11,12]). However, high costs and coarse spatial and temporal resolution of many remote sensing products hamper the investigation of highly variable and dynamic glaciological processes (e.g., [13,14]). To overcome some of the drawbacks related to satellite remote sensing, Unmanned Aerial Vehicles (UAVs) have been increasingly deployed in glaciological research [14].

To date, most of the UAV surveys in glaciology have been performed in the polar and subpolar regions [14]. Despite the limited accessibility of alpine glaciers and the challenges related to flying in high-mountain environments (complex terrain, low air pressure, poor reception of GPS satellite signals, etc.), the suitability of UAVs for high-resolution glacier monitoring has been successfully demonstrated in the Alps [15–18] and the Himalaya [19–22] in recent years.

Apart from a few studies (e.g., [23,24]), most of the glaciological mapping and monitoring campaigns use off-the-shelf UAVs to acquire aerial images in a high spatial resolution (e.g., [15,18,19]). The benefit of commercial UAVs is obvious: they are reliable and ready-to-use. However, high purchase costs for commercial UAVs and proprietary photogrammetry software are hardly affordable within the budget of smaller projects. Furthermore, commercial UAVs may not be the first choice in harsh environments, where the potential damage or loss of scientific equipment is an issue. Self-developed fixed-wing UAVs equipped with optical or meteorological sensors are a low-cost alternative but have been tested on ice sheets [23–26] rather than on alpine glaciers.

To assess the potential of autonomous low-cost UAVs for high-resolution mapping and monitoring of alpine glaciers, we conducted multiple surveys on the Kanderfirn in the Swiss Alps during the ablation seasons in 2017 and 2018 using technology of the complete open-source hardware and software project Paparazzi UAV [27]. For the generation of high-resolution orthophotos and digital surface models (DSMs) from aerial imagery, we tested the open-source software OpenDroneMap [28]. Since this is the first study using OpenDroneMap for applications in the field of glaciology and geomorphology, we cross-checked the results with outputs of the well-established proprietary photogrammetry software Pix4D [29].

## 2. Study Site

The Kanderfirn (46.48° N, 7.80° E), where the UAV surveys were performed, is a south-west-facing valley glacier in the Bernese Alps in western Switzerland (Figure 1). The glacier is located in the upper catchment of the Kander River and bounded by the Blüemlisalp Massif (3661 m a.s.l.) in the north, the Mutthorn (3038 m a.s.l.) in the east and the Petersgrat (3202 m a.s.l.) in the south. The lower part of the glacier tongue is also known as Alpetli Glacier. The main accumulation area stretches from the Petersgrat to the western side of the Tschingelhorn (3562 m a.s.l.). In 1850, at the end of the Little Ice Age, the Kanderfirn covered an area of ca. 16.0 km$^2$ [30,31]. Since then, the area has continuously decreased to 13.8 km$^2$ in 1973 [30–32], 12.2 km$^2$ in 2010 [33], and 11.2 km$^2$ in 2018. Between 1973 and 2018, the glacier terminus retreated laterally by more than 500 m [34] and it is now located at ~2330 m a.s.l. (Figure A1). The ice volume stored in the glacier was in the order of 1.39 ± 0.36 km$^3$ in 1999 [35] but substantially decreased afterwards. Helicopter-borne ground-penetrating radar investigations for bedrock mapping have not yet been completed at the Kanderfirn [36], but model results indicate that the ice is on average 90 ± 30 m and at the maximum 250 ± 75 m thick [37]. A long-term mass balance time series, as is available for other Swiss glaciers, does not exist for the Kanderfirn [38]. An automatic weather station of the Institute for Snow and Avalanche Research (SLF) is located 7.5 km west of the Kanderfirn at the Fisistock (46.47° N, 7.67° E, 2155 m a.s.l.). Hourly meteorological data of these stations are provided by MeteoSwiss. For the period 2002–2018, the measured mean annual air temperature was 2.8 °C and the mean annual precipitation 826 mm. The mean maximum snow height at the end of each winter season was 2064 mm.

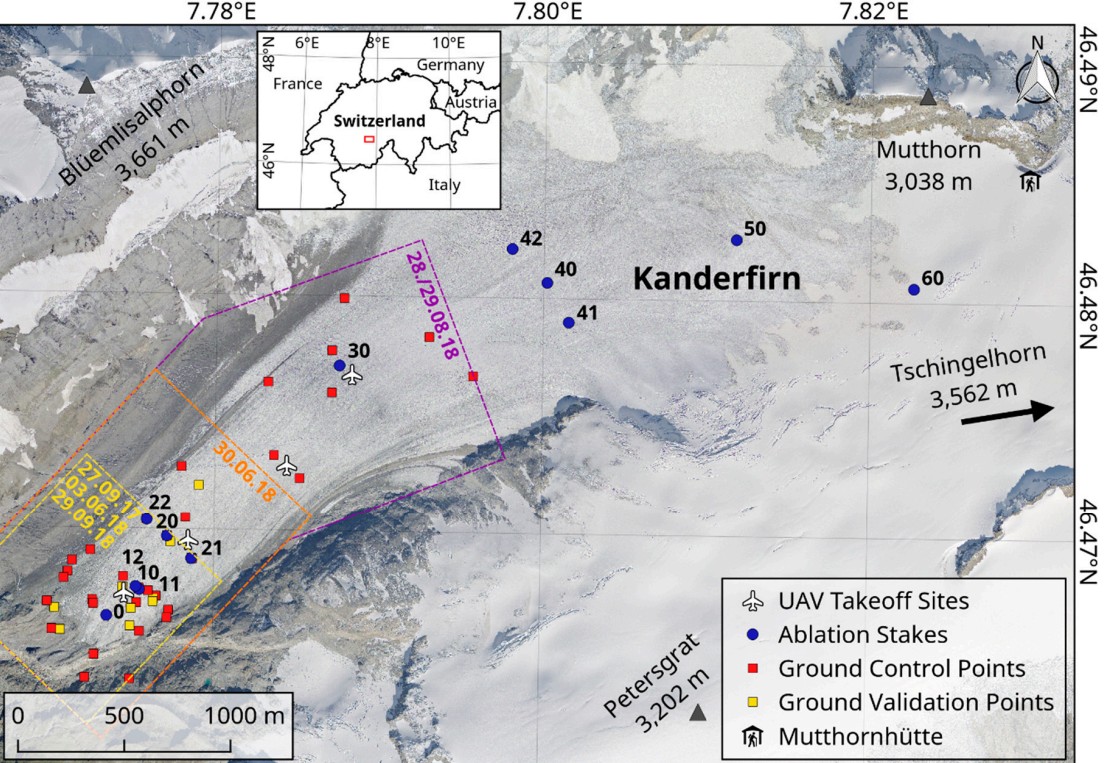

**Figure 1.** Overview map of the Unmanned Aerial Vehicle (UAV) test site and glacier monitoring network on the Kanderfirn in the Bernese Alps (Switzerland). The dashed outlines (yellow, orange and purple) indicate the spatial coverage of the different UAV surveys in 2017 and 2018. The background image is a high-resolution (25 cm) orthomosaic from 2014 provided by the Swiss Federal Office of Topography.

The Kanderfirn was chosen for the UAV surveys and glacier monitoring since it fulfils the basic requirements (area >2 km$^2$, altitude range >500 m, simple geometry, smooth surface, well-defined catchment area, etc.) for spatially representative glacier surface mass balance investigations [39].

## 3. Materials and Methods

### 3.1. Unmanned Aircraft System

For the implementation of autonomous aerial surveys at the Kanderfirn, we designed a low-cost Unmanned Aircraft System (UAS) based on hardware and software of the complete open-source project Paparazzi UAV. As every UAS, it consists of an airborne, a ground, and a communication segment [27].

Any usual model aircraft, either self-built or as an off-the-shelf product, can be utilised as UAV. We built a flying wing from expanded polypropylene (EPP) fuselage parts (model Knurrus Maximus FPV) with a wingspan of 140 cm (Figure 2). In contrast to other model types, flying wings are simple to build and very robust concerning gusts and mechanical damage. A brushless driver motor (NTM Prop Drive Series 35-42A) was mounted at the back to work as a pusher with 12- × 6-inch folding carbon propellers, providing considerable thrust also at low rates of rotation. The motor is actuated by a speed controller (Turnigy Plush 60A). Two servomotors (Multiplex Hitec Digital Servos) were installed for actuating the aileron rudders. The rudders are used to fly curves by changing the roll attitude and to alter the flight level by changing the pitch. Usually, the speed controller and servomotors are controlled by a remote control (rc) receiver directly. However, for autonomous flying, an autopilot controller (Apogee) was plugged between rc-receiver and the motors. The Apogee device was developed by ENAC (Ecole Nationale de l'Aviation Civile in France) and is available as open hardware within the Paparazzi project. It is equipped with a processor (ARM STM32F405RGT6 Cortex M4), an inertial measurement unit (IMU), a barometer and a SD-card slot. Furthermore, it offers a number of connectors,

e.g., for plugging the rc-receiver, motor controllers, GPS module, telemetry modem and any other kind of sensor supported by SPI (Serial Peripheral Interface), I²C (Inter-Integrated Circuit) or UART (Universal Asynchronous Receiver Transmitter) protocol. An 11.1 V lithium polymer battery with 5000 mAh powers the whole system and enables flight times of up to 45 min, mainly depending on the vertical distance to climb. Since the payload of the UAV is limited to ca. 250 g, we equipped it with a lightweight 12-megapixel digital camera (GoPro Hero 5 Black, weight: 120 g, lens focal length: 3.0 mm, sensor dimensions: 6.17 × 3.47 mm, resolution: 4000 × 3000 pixel). The take-off-weight of the UAV (including camera and battery) was less than 2 kg.

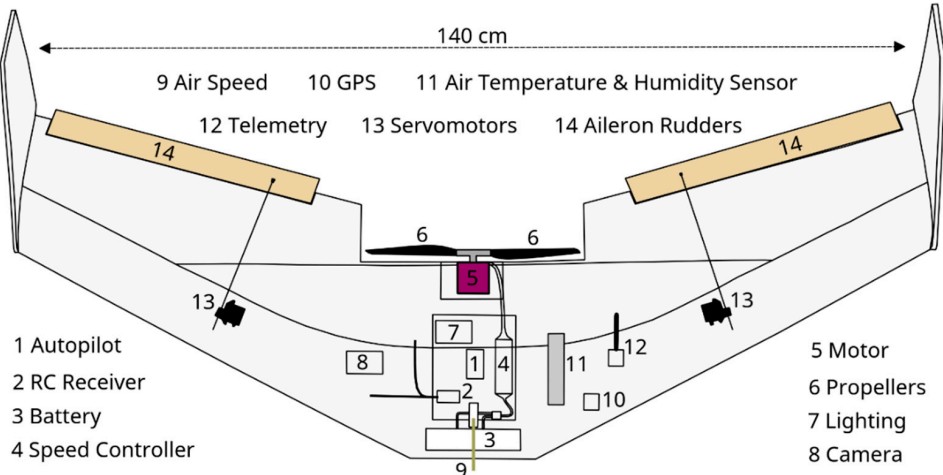

**Figure 2.** Setup of the low-cost UAV (flying wing model).

Flight planning and monitoring was accomplished using the Paparazzi software environment, which provides a graphical user interface (Paparazzi Center) to program and configure the autopilot controller firmware and user-specific flight plans. After flashing the firmware and flight plan, the UAV was ready for take-off once all pre-flight-checks were successful. Paparazzi supports three different flying modes: manual flying by remote control, no intervention of the autopilot ("manual"); manual flying by remote control, but autopilot is assisting for stabilisation of the UAV ("auto1"); autonomous flying by executing a predefined flight plan ("auto2"). The autonomous flight can be simulated and monitored in Paparazzi using the ground control station's (GCS) graphical interface (Figure 3).

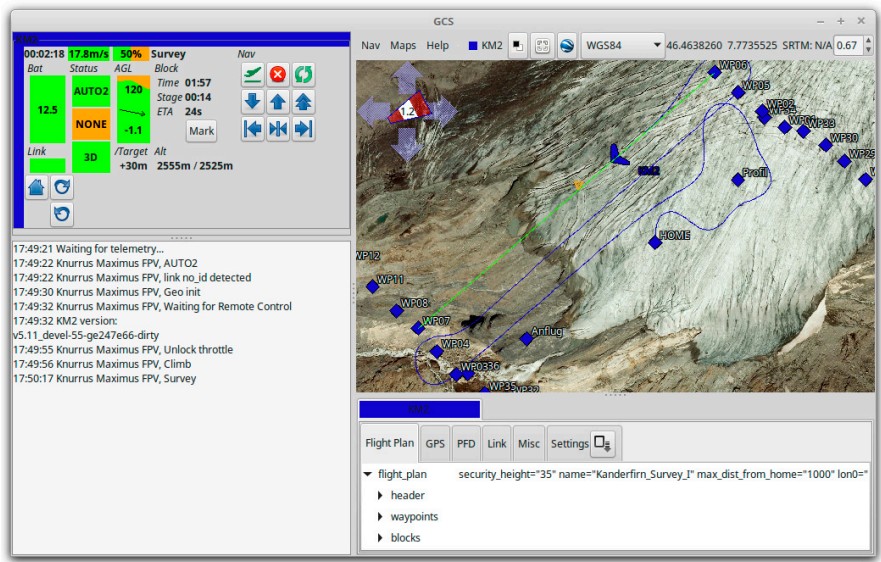

**Figure 3.** Graphical user interface of the Paparazzi ground control station (GCS).

The communication from ground with the UAV is ensured in two ways during the autonomous flight: firstly by remote control as a permanent backup and secondly through a bidirectional telemetry link (Xbee 2.4 GHz modems). The telemetry link serves for transmitting the recent position and state of the UAV and its sensors to the GCS. Conversely, it also allows control of the UAV by commands executed in the GCS. In this way, the flight plan or certain parameters of it (e.g., throttle) can be modified in real time. Flight information data were recorded on the GCS computer and in high frequency on the SD-card on board. The entire UAS, including camera and remote control, but without a GCS computer, costs about 1300 € (an overview of the individual components and cost estimate of the UAS is given in Table A1).

## 3.2. Aerial Surveys and In-Situ Measurements

We conducted a first photogrammetric test survey over the forefield and terminus of the Kanderfirn at the very end of the ablation season of 2017 using the open-source UAS setup described above (Section 3.1). However, we did not perform any ground control measurements on that day. Between June and October 2018, we visited the Kanderfirn twice a month to monitor the glacier surface evolution over the melt period. Aerial photographs of the ablation area were acquired with the introduced fixed-wing UAV during nine flights on five different days (Table 1). The UAV automatically followed a pre-defined flight path, which was created with the Paparazzi UAV mission planning software and uploaded to the autopilot beforehand. Only take-off and landing were manually operated. Each survey lasted about 15 min and covered an area of ca. 0.8 km$^2$. Two adjacent surveys were conducted at the end of June and four adjacent surveys at the end of August to cover a larger area of the glacier tongue (Figure 1). The last two surveys of the melt season (29 September 2018, flight no. 1 and 2) were aborted before completion due to unexpected technical problems.

**Table 1.** Characteristics of the ten UAV surveys performed at the Kanderfirn in 2017 and 2018.

| Date | Flight No. | Start Time (hh:mm) | Flight Time (hh:mm) | Flight Altitude (m a.g.l.) | Area (km$^2$) | Images (selected) | Resolution (cm/pixel) |
|---|---|---|---|---|---|---|---|
| 27 September 2017 | 1 | 16:26 | 00:14 | 140 ± 10 | 0.7 | 1242 (314) | 7.2 ± 0.5 |
| 3 June 2018 | 1 | 14:37 | 00:16 | 140 ± 10 | 0.7 | 913 (347) | 7.2 ± 0.5 |
| 30 June 2018 | 1 | 15:03 | 00:16 | 120 ± 10 | 0.8 | 972 (249) | 6.2 ± 0.5 |
| 30 June 2018 | 2 | 18:02 | 00:16 | 135 ± 20 | 0.8 | 952 (228) | 6.9 ± 1.0 |
| 28 August 2018 | 1 | 13:27 | 00:15 | 120 ± 10 | 0.8 | 883 (210) | 6.2 ± 0.5 |
| 28 August 2018 | 2 | 15:24 | 00:16 | 135 ± 20 | 0.8 | 935 (210) | 6.9 ± 1.0 |
| 28 August 2018 | 3 | 17:14 | 00:17 | 135 ± 20 | 0.8 | 992 (217) | 6.9 ± 1.0 |
| 29 August 2018 | 1 | 12:20 | 00:17 | 135 ± 20 | 0.8 | 1036 (213) | 6.9 ± 1.0 |
| 29 September 2018 | 1 | 10:51 | 00:11 | 120 ± 10 | 0.8 | 668 (215) | 6.2 ± 0.5 |
| 29 September 2018 | 2 | 16:15 | 00:01 | 120 ± 10 | <0.1 | 70 (0) | 6.2 ± 0.5 |

To correct for the barrel distortion associated with wide-angle lenses, we chose the linear-field-of-view mode available in the camera settings. This mode applies an internal algorithm to correct for lens distortion before saving the image [40]. At an average flight altitude of 135 ± 20 m above ground level, the camera took photos with a pixel resolution of 5.7–7.7 cm.

For accurate georeferencing and validation of the final remote sensing products (Section 3.3), we distributed ground control points (GCPs) across the ablation zone and glacier forefield. We placed white Teflon markers (A2 paper size) on bedrock and red Teflon markers (same size) on ice or snow (Figure 4). The position of the centre of each GCP was accurately measured with a Trimble Geo 7X handheld differential global navigation satellite system (dGNSS) [41].

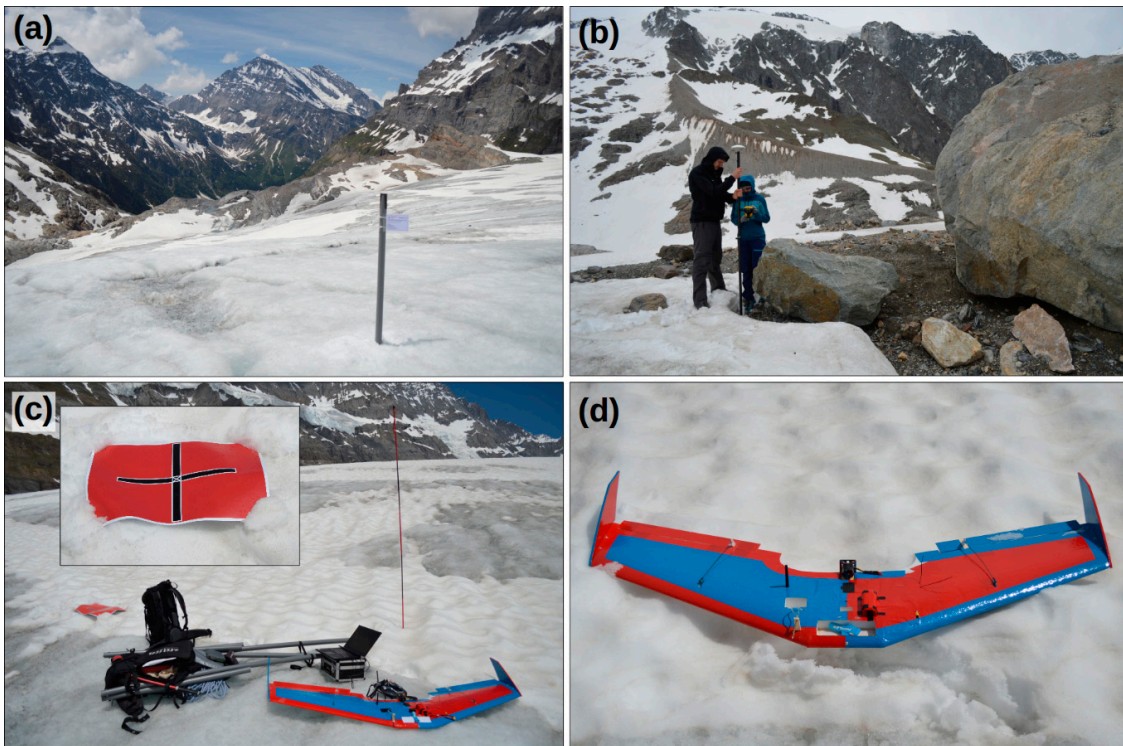

**Figure 4.** Fieldwork at the Kanderfirn. (**a**) Ablation stake (no. 10). (**b**) dGNSS measurement of a natural ground control points (GCP) in the glacier forefield. (**c**) Teflon marker (A2 paper size) and GCS computer. (**d**) Low-cost UAV.

As an independent measure for the spatiotemporal variability of ice melt rates across the ablation zone, we installed 13 ablation stakes along the flow line between 2365 and 2843 m a.s.l. on the Kanderfirn (Figure 1, Table 2). The position and glacier surface elevation change at each ablation stake was regularly measured with the aforementioned dGNSS-receiver. The ablation (in cm) was obtained directly by subtracting stake readings from different visiting dates.

**Table 2.** Overview of the ablation measurements at the Kanderfirn between June and October 2018.

| Stake | Lat (°N) | Lon (°E) | Elevation (m) | Start Date | End Date | Period (d) | Ablation (cm) | Ablation (cm d$^{-1}$) |
|---|---|---|---|---|---|---|---|---|
| 00 | 46.4663 | 7.7735 | 2363 | 30 June 2018 13:00 | 23 October 2018 11:40 | 114.9 | 549 | 4.8 |
| 10 | 46.4675 | 7.7754 | 2414 | 3 June 2018 15:00 | 23 October 2018 11:30 | 141.9 | 648 | 4.6 |
| 11 | 46.4674 | 7.7755 | 2413 | 3 June 2018 15:00 | 23 October 2018 11:25 | 141.9 | 610 | 4.3 |
| 12 | 46.4675 | 7.7752 | 2413 | 30 June 2018 11:50 | 23 October 2018 11:35 | 115.0 | 521 | 4.5 |
| 20 | 46.4697 | 7.7771 | 2444 | 30 June 2018 16:50 | 23 October 2018 11:00 | 114.8 | 443 | 3.9 |
| 21 | 46.4688 | 7.7786 | 2437 | 30 June 2018 16:15 | 23 October 2018 11:15 | 114.8 | 489 | 4.3 |
| 22 | 46.4704 | 7.7759 | 2446 | 30 June 2018 17:10 | 23 October 2018 11:05 | 114.7 | 509 | 4.4 |
| 30 | 46.4770 | 7.7875 | 2544 | 24 July 2018 14:15 | 23 October 2018 10:20 | 90.8 | 347 | 3.8 |
| 40 | 46.4807 | 7.8002 | 2633 | 8 August 2018 15:15 | 23 October 2018 09:50 | 75.8 | 204 | 2.7 |
| 41 | 46.4790 | 7.8016 | 2632 | 24 July 2018 15:30 | 23 October 2018 00:00 | 90.4 | 335 | 3.7 |
| 42 | 46.4821 | 7.7980 | 2641 | 8 August 2018 15:45 | 23 October 2018 09:45 | 75.8 | 284 | 3.7 |
| 50 | 46.4826 | 7.8118 | 2735 | 8 August 2018 16:30 | 23 October 2018 09:30 | 75.7 | 186 | 2.5 |
| 60 | 46.4806 | 7.8227 | 2843 | 9 August 2018 09:30 | 23 October 2018 09:00 | 75.0 | 136 | 1.8 |

### 3.3. Generation of Orthophotos and DSMs

We derived five high-resolution orthophotos (5 cm) and DSMs (25 cm) of the Kanderfirn from UAV-based aerial images using the latest version (0.4.1) of the photogrammetry software OpenDroneMap (www.opendronemap.org). OpenDroneMap is a rapidly evolving community-based open-source toolkit for processing and analysing aerial imagery acquired with UAVs [28]. The command line program runs on all major operating systems. Different extendable web applications, such as WebODM, provide an interface to OpenDroneMap and enable data visualisation, storage and analysis [42]. OpenDroneMap relies on the following photogrammetry workflow to generate dense point clouds, textured meshes, DSMs and orthophotos from aerial imagery (Figure 5):

1.  Import of (geotagged) aerial images and extraction of image metadata (camera specifications and geographical information).
2.  Calculation of accurate camera positions/orientations and generation of a sparse three-dimensional (3D) point cloud using the structure from motion library OpenSfM that performs feature extraction and matching [43].
3.  Densification of sparse point cloud based on Multi-View Stereo 3D reconstructions [44].
4.  Conversion of dense point cloud into a triangular 3D mesh based on an implemented Poisson Surface Reconstruction [45].
5.  Texturing of 3D mesh using an algorithm for large-scale 3D reconstructions. As data input, the algorithm requires a triangulated 3D mesh and images that are registered against this model [46].
6.  Georeferencing of 3D point cloud and triangular mesh. An affine transformation with three GCPs is applied to align the 3D models. For the affine transformation, OpenDroneMap chooses a combination of three GCPs that yields the highest possible accuracy.
7.  Generation of a georeferenced DSM from the dense point cloud.
8.  Generation of a georeferenced orthophoto from the textured mesh.

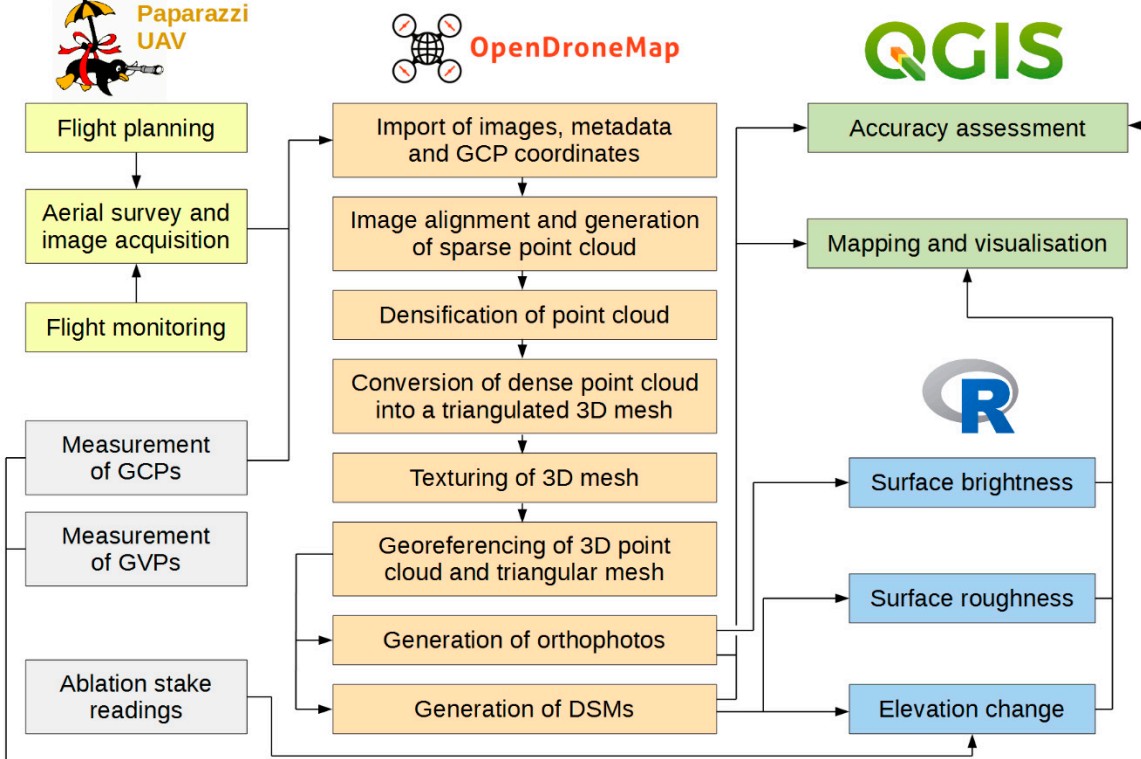

**Figure 5.** Workflow for the generation and analysis of UAV-based orthophotos and digital surface models using open-source software only.

We selected between 200 and 350 images from the complete dataset of each survey to generate orthophotos and DSMs (Table 1). Images acquired during the take-off, landing and hairpin turns were removed from the selection to exclude all low-altitude and non-nadir observations. The orthophotos and DSMs from 27 September 2017, 3 June 2018 and 29 September 2018 were produced with images from only one survey, covering solely the glacier terminus and forefield. For the larger orthophotos and DSMs from 30 June and 28–29 August 2018, images of two and four adjacent surveys, respectively, were batched. All orthophotos and DSMs were finally clipped to a similar geographic extent.

For the accurate georeferencing of orthophotos and DSMs, OpenDroneMap requires a GCP file. Such a file can be created and edited using a simple text editor or the GCP interface available in WebODM. The GCP file needs to include the GPS position (X, Y and Z coordinates) of at least five GCPs that are visible on three or more aerial images [47]. We manually searched for GCPs on the aerial images that were obtained near the respective measurement site. Due to the flight level and the strong surface reflection of the surrounding ice or snow, not all GCPs distributed on the glacier were found again on the images. About two thirds of the detected GCPs were considered for the georeferencing process in OpenDroneMap, while the others served as ground validation points (GVPs). In the case of the test survey from 27 September 2017, where no GCPs were placed on the glacier, we relied on the measured GCP locations from 3 June 2018. This approach neglects glacier dynamics and the potential displacement of the GCP locations over time, but due to small surface velocities and insignificant ablation during the winter season, we assumed that the resulting accuracy was still higher than the uncertainty (several meters) of the camera's internal GPS.

*3.4. Calculation of Surface Brightness, Roughness and Elevation Changes*

The surface albedo of glaciers varies considerably in space and time and is an important variable to monitor since it largely determines the spatial and temporal heterogeneity of melt rates across the ablation zone (e.g., [48]). Calibrated high-resolution albedo products can be obtained using raw images of a lightweight consumer-grade digital camera installed on a UAV if information on the upward and downward radiation at the study site are available [49]. Since no additional radiometric measurements were performed at the Kanderfirn, we used the surface brightness of the glacier as a rough proxy for the broadband surface albedo [17]. This approach assumes a linear relationship between RGB values (signal recorded by the digital camera) and the surface reflectance of the glacier under similar illumination conditions [17,50]. To ensure that RGB values remain proportional to the surface brightness, aerial images were acquired with a fixed white balance [49]. We determined the surface brightness for each orthophoto by calculating the arithmetic mean of each pixel's RGB value [18].

Besides albedo, surface roughness is another key parameter affecting the surface energy balance of glaciers [51,52]. To derive the surface roughness from topographic data, we followed the method proposed by Rippin et al. [17]. We detrended the DSMs to ensure that the large-scale topography had no impact on the roughness calculation [53]. For this purpose, we smoothed the DSM using a moving window of 5.25 m. A window size of 5.25 m was chosen since this was found to be the smallest window size, resulting in a detrended surface with a mean close to zero. The smoothed DSM was then subtracted from the original DSM to obtain a residual that exhibited small-scale surface features in the order of a few decimetres to meters. To capture the vertical variation of the topography within a certain area, we used the standard deviation as a proxy for surface roughness [17]. The standard deviation was calculated for a set of moving windows in the range of 5.25 to 20.25 m.

For quantifying surface elevation changes over the glacier area during the ablation season of 2018, we calculated the elevation difference by subtracting the georeferenced DSMs from 3 June and 29 September (e.g., [16,18,19]). Furthermore, we compared the OpenDroneMap DSM from August 2018 with the swissALTI3D from 2010 (for more information regarding the product, see Section 3.5) to study the long-term changes of the glacier surface. To be independent of the varying extent of the

orthophotos and DSMs from different surveys, we analysed the evolution of the surface elevation, brightness and roughness over the melt period within a predefined comparative area.

### 3.5. Quality Assessment

We assessed the accuracy of the OpenDroneMap orthophotos and DSMs using ground reference measurements from the ablation season of 2018 (Section 3.2). On each orthophoto, the horizontal (XY) distance between the visible location and actual position of every GCP and GVP was determined manually using the open-source geographic information system QGIS [54]. In accordance with this procedure, we also measured the XY displacement of nine prominent objects in the glacier forefield that are visible on the orthophotos, but also on the SwissImage from 2014 (background image in Figure 1). The SwissImage is a national orthomosaic with a spatial resolution and accuracy of 25 cm, provided by the Swiss Federal Office of Topography [55]. As a measure for the overall accuracy of orthophotos, we calculated the Root Mean Square Error (RMSE) from the individual XY displacements.

To check whether the orthophotos and corresponding DSMs were precisely aligned, we generated a hillshade from each DSM and visually compared the 3D product with the respective orthophoto using distinct surface features (e.g., cliffs, crevasses, supraglacial dirt cones) as references. Since no displacement was observed, we assumed that the XY accuracy stated for the orthophotos was also valid for the DSMs. To assess the vertical accuracy of each DSM, we subtracted the measured elevation of every GCP or GVP from the corresponding pixel value of the DSM. In addition, we calculated the pixel difference between the DSMs and swissALTI3D (image acquisition: 2010) within one reference rectangle outside the glacier area. The swissALTI3D is a national DSM provided by the Swiss Federal Office of Topography and has a spatial resolution of 2 m. For areas above 2000 m, the mean XYZ accuracy of the swissALTI3D is 1–3 m [56]. The overall vertical (Z) accuracy of each OpenDroneMap DSM is stated again as RMSE of the individual displacements at the GCPs/GVPs and within the reference area.

Besides the quality of the aerial images, the number of GCPs, the accuracy of dGNSS measurements, etc. [57], the quality and accuracy of the orthophotos and DSMs is subject to the applied photogrammetry software. Since OpenDroneMap has not been extensively evaluated yet, we also computed orthophotos and DSMs for the beginning and end of the ablation season 2018 using the well-established photogrammetry software Pix4D [29]. To assess the differences of both software products, we compared the overall accuracy (RMSE) and quality (e.g., surface texture) of the different datasets.

In theory, the residual of the DSM difference and measured ablation at the stakes between two different dates can be explained by ice dynamics if the combined error of both methods is much smaller than the actual ice dynamic. To test whether the accuracy of the OpenDroneMap DSMs is sufficient to quantify the ice dynamics at the location of the stake measurements, we compared the DSM difference, GPS difference and ablation at stakes no. 00, 10, 11, 12, 20, 21 and 22 (Table 2) over the melt season of 2018.

## 4. Results

### 4.1. Performance of UAV

Eight aerial surveys for the generation of high-resolution orthophotos and DSMs were successfully accomplished with the developed low-cost UAV at the Kanderfirn in 2017 and 2018 (Table 1). The last two surveys of the season (29 September 2018) were aborted before completion due to a defective servomotor. However, the surveys still provided enough images to produce a gapless orthophoto and DSM of the glacier tongue for this day. Apart from the last flight, each survey covered an area of about 0.8 km$^2$. Longer flight times and larger surveys were possible with the 5000 mAh battery (the remaining capacity after landing was still >40%), but the limited visibility of the UAV restricted the maximum flight distance. Despite the intense colours chosen for the lamination of the wings (Figure 4), visual monitoring of the UAV became difficult when the distance to the GCS exceeded 500 m. Since

the visibility and manual control of the UAV is required by law and needs to be guaranteed for safety reasons (e.g., immediate landing in case of an approaching helicopter), the maximum flying distance had to be limited to 500 m. For other locations, the maximum flying distance might be adjusted depending on the legal regulations and weather conditions.

The used fixed-wing model and chosen hardware configuration (Table A1) proved to be suitable for aerial surveys in alpine environments. The reception and accuracy (few meters) of the GPS signal in the mountains was sufficient for autonomous flying. Local winds (e.g., mountain/valley breeze) and gusts did not substantially affect the flying dynamics of the UAV due to flight stabilisation through the autopilot controller. The UAV was also able to withstand hard landings on the glacier surface. All electronic parts were well protected by the surrounding EPP and did not experience any damage. In cases where the EPP fuselage was slightly ripped, it could be easily repaired.

The communication from ground with the UAV by remote control functioned reliably during all surveys. In addition, we could also track the flight path of the UAV on the Paparazzi GCS screen. The bidirectional 2.4 GHz telemetry link between the GCS and UAV was stable except for some shorter interruptions and allowed the monitoring of the survey in real time.

### 4.2. Accuracy of Orthophotos and DSMs

We obtained a horizontal (XY) and vertical (Z) accuracy for the measured GCPs and GVPs of <0.1 m after postprocessing the dGNSS data. The mean XY accuracy (RMSE) of the OpenDroneMap orthophotos largely depended on the size of the covered area and varied from 0.7 m for smaller orthophotos to 1.2 m for larger orthophotos (Table 3). However, the XY displacement at the individual GCPs and GVPs can be much larger or smaller than the actual RMSE (Table S1). The vertical accuracy of the OpenDroneMap DSMs ranged from 0.7 to 2.1 m. Compared to the OpenDroneMap results, the Pix4D orthophotos (XY RMSE of 0.3–0.5 m) and DSMs (Z RMSE of 0.4–0.5 m) are more accurate (Table 3).

**Table 3.** Root mean square error (RMSE) of the orthophotos (XY) and DSMs (Z) generated with OpenDroneMap and Pix4D based on the evaluation of all available GCPs and GVPs.

| Date | Software | Version | GCPs | GVPs | XY RMSE (m) | | | Z RMSE (m) | | |
|---|---|---|---|---|---|---|---|---|---|---|
| | | | | | GCP | GVP | Total | GCP | GVP | Total |
| 27 September 2017 | ODM | 0.4.1 | 5 | 4 | 1.3 | 1.1 | 1.2 | 1.0 | 0.6 | 0.9 |
| 3 June 2018 | ODM | 0.4.1 | 5 | 3 | 0.6 | 0.7 | 0.7 | 0.7 | 0.8 | 0.7 |
| 3 June 2018 | Pix4D | 4.3.31 | 5 | 3 | 0.3 | 0.7 | 0.5 | 0.3 | 0.3 | 0.4 |
| 30 June 2018 | ODM | 0.4.1 | 9 | 3 | 1.5 | 0.4 | 1.2 | 2.3 | 1.2 | 2.1 |
| 28./29 August 2018 | ODM | 0.4.1 | 22 | 5 | 1.3 | 0.9 | 1.2 | 2.1 | 0.9 | 1.9 |
| 29 September 2018 | ODM | 0.4.1 | 6 | 4 | 0.6 | 0.7 | 0.7 | 0.9 | 0.9 | 0.9 |
| 29 September 2018 | Pix4D | 4.3.31 | 6 | 4 | 0.2 | 0.4 | 0.3 | 0.2 | 0.7 | 0.5 |

Apart from two exceptions, the XY displacement between the UAV orthophotos from 2017/2018 and the SwissImage from 2014 is ≤0.9 m (RMSE) at the reference points in the glacier forefield and therefore larger than the XY accuracy (0.25 m) of the SwissImage itself (Table A2). In contrast, the Z displacement between the UAV DSMs from 2017/2018, and swissALTI3D from 2010 within the reference area in the glacier forefield is much larger and ranges from a RMSE of 0.6 to 3.4 m (Table A2).

The two orthophotos produced with OpenDroneMap and Pix4D are in good agreement (Figure 6). Data gaps and artefacts (e.g. blurry or patchy areas) resulting from failed or erroneous image alignments were mainly restricted to the edges in both products due to a lack of overlapping images. Even though the same aerial images were selected for the generation of orthophotos, the OpenDroneMap products covered a larger area than the Pix4D products from the same day. Furthermore, abrupt transitions in brightness due to variations in cloudiness during image acquisition are only visible on the Pix4D orthophoto from 29 September 2018 and not on the corresponding OpenDroneMap orthophoto (Figure 6). Distinct differences between both products become apparent when comparing the different

DSMs from 3 June and 29 September 2018 (Figure 6). Both, the OpenDroneMap and Pix4D DSM comparisons, reveal a general surface lowering across the glacier tongue over the ablation season. However, the OpenDroneMap and the Pix4D DSMs from the same day differed by <1 m in the centre to >3–5 m at the edges (Figure 6).

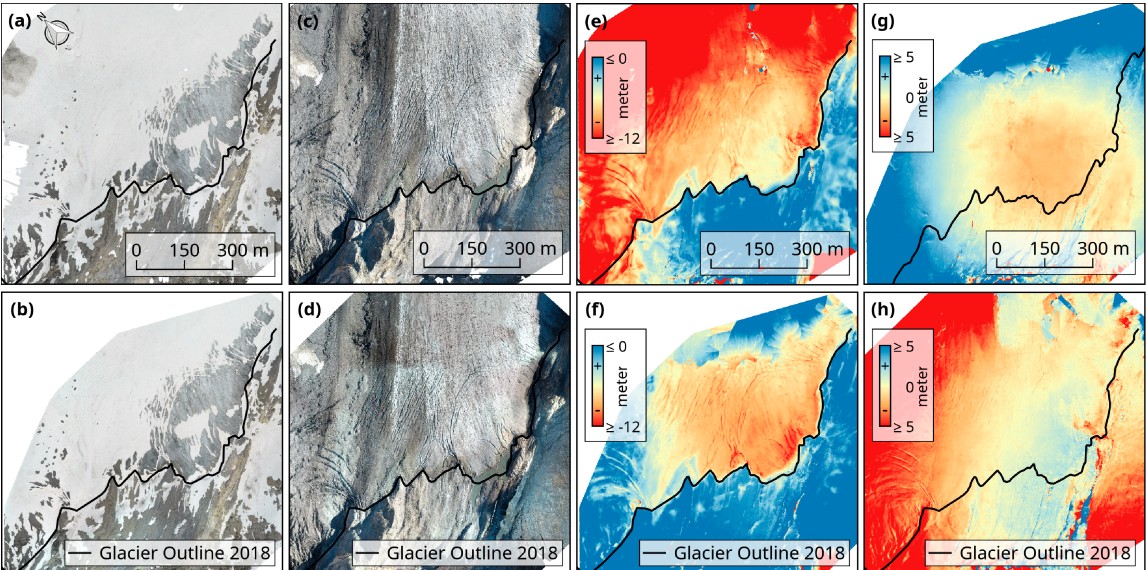

**Figure 6.** Comparison of OpenDroneMap and Pix4D outputs. (**a**) OpenDroneMap and (**b**) Pix4D orthophotos from 3 June 2018. (**c**) OpenDroneMap and (**d**) Pix4D orthophotos from 29 September 2018. (**e**) Elevation change over the melt season 2018 based on the OpenDroneMap DSM difference between 3 June and 29 September. (**f**) Elevation change over the melt season 2018 based on the Pix4D DSM difference between 3 June and 29 September. (**g**) OpenDroneMap and Pix4D DSM difference from 3 June and (**h**) 29 September 2018, respectively.

Despite the spatial discrepancies between the OpenDroneMap and Pix4D DSMs, the pixel-based elevation changes over the melt season 2018 derived from subtraction of the consecutive OpenDroneMap and Pix4D DSMs (29.09. minus 03.06.) both followed a bimodal distribution (Figure 7). The two elevation change maxima in both datasets are located at 0 and −8 m. The first maximum at 0 m represents the stable terrain outside the glacier area, whereas the second maximum at −8 m indicates the mean surface lowering across the glacier tongue over the melt season. However, the DSM-based elevation changes differ from the measured melting and surface lowering at the ablation stakes by ±0.4–2.7 m (Table 2 and Table S2). Furthermore, the subtraction of the OpenDroneMap DSMs yields unrealistic elevation changes of >−10 m, which indicate a distortion towards the edges of the DSMs.

### 4.3. Glacier Surface Changes

Five high-resolution orthophotos (5 cm pixel resolution) and DSMs (25 cm pixel resolution) of the Kanderfirn were successfully obtained using OpenDroneMap and aerial images from nine UAV surveys in 2017 and 2018. The orthophotos and DSMs allowed for detailed geomorphological mapping and facilitated the analysis of spatial changes of the glacier surface over the melt season. Rapid small-scale variations like downwasting of the glacier tongue, formation of proglacial lakes, heterogeneous disappearance of winter snow and disintegration of avalanche cones below the Blüemlisalp south face were well captured by the consecutive orthophotos (Figure 8).

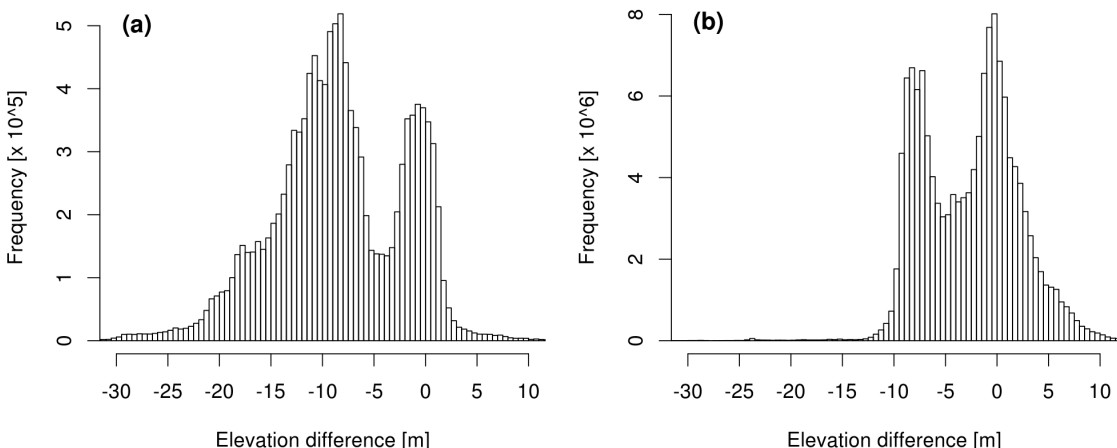

**Figure 7.** Frequency distribution of pixel values derived from the difference of the DSMs from 3 June and 29 September 2019. (**a**) Histogram of the OpenDroneMap DSM difference. (**b**) Histogram of the Pix4D DSM difference.

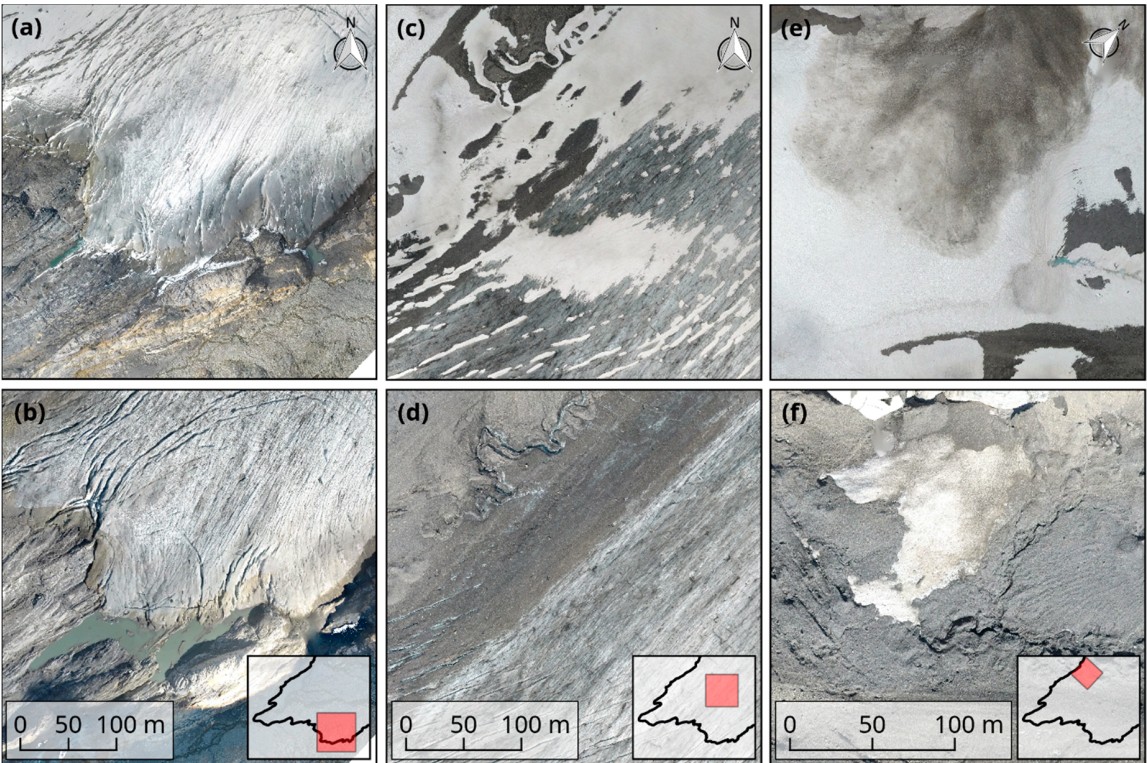

**Figure 8.** Three examples of rapid changes of the glacier tongue over the melt season of 2018. Downwasting of the glacier tongue and formation of a proglacial lake between 27 September 2017 (**a**) and 29 September 2018 (**b**). The disappearance of the supraglacial snow cover between 30 June 2018 (**c**) and 28 August 2018 (**d**). The disintegration of an avalanche cone between 30 June 2018 (**e**) and 29 September 2018 (**f**).

General variations of the surface texture like changes in brightness or roughness are also visible on the orthophotos and DSMs (Figure 9). At the end of the melt season of 2017, a thin snow layer covered the glacier tongue, but crevasses and supraglacial meltwater channels were still visible. At the beginning of the melt season of 2018, most of the terminus was covered by snow. Accordingly, the surface was very smooth. Between the UAV acquisitions at the end of June and at the end of August, the remaining snow melted away and the underlying ice surface became exposed. At that time, the sharp contrast between the debris-covered area below the Blüemlisalp Massif and remaining

bare ice glacier surface was evident. At the end of the ablation season, the glacier surface became very rough due to the continuous widening of crevasses and deepening of supraglacial meltwater channels.

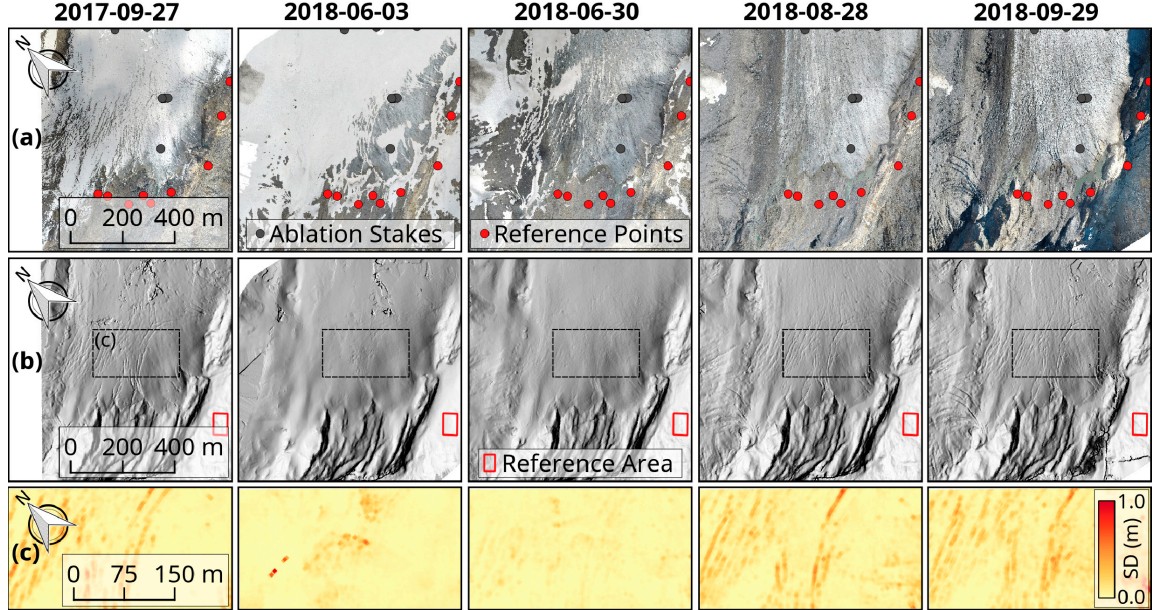

**Figure 9.** Spatial and temporal changes of the studied glacier surface. (**a**) Orthophotos (5 cm pixel resolution) of the tongue of the Kanderfirn generated with OpenDroneMap using images from nine UAV surveys performed on five different days in 2017 and 2018. (**b**) OpenDroneMap DSMs (25 cm pixel resolution) from the same days. (**c**) The variation of the surface roughness (here defined as standard deviation of the detrended DSM within a $5.25 \times 5.25$ m moving window) of the glacier tongue over the ablation season.

The ice melt at the ablation stakes in summer 2018 varied from 1.4 m at 2843 m a.s.l. to 6.5 m at the terminus (Table 2). There, ablation took place from the beginning of June until the end of October. The glacier surface lowering determined with dGNSS at the location of the stakes followed a similar trend but was slightly larger compared to the measured ablation due to the general movement of the stakes towards lower elevations (Figure 10). The areal surface elevation differences derived from the consecutive OpenDroneMap DSMs are comparable to the dGNSS point measurements, but overestimate the total surface lowering at the end of the season by 0.6 m. In contrast to the brightness of the glacier surface, which was derived from the orthophotos and decreased linearly over the melt season (from a mean RGB value of 165 to 142), the surface roughness derived from the DSMs increased steadily (from 6 to 13 cm). All the tested moving window sizes between 5.25 and 20.25 m captured the spatial and temporal variations of surface roughness. However, smaller windows sizes were prone to extreme values, while larger window sizes did not capture narrow crevasses or other small-scale features.

Larger orthophotos and DSMs can be generated with OpenDroneMap by combining aerial images from multiple adjacent surveys (Figure 11). We obtained two orthophotos and DSMs of the Kanderfirn for 30 June based on 474 images from two surveys and for 28–29 August based on 850 images from four surveys. The June orthophoto and DSM covered an area of 1.7 km² (15%), while those from August covered an area of 3.4 km² (30% of the total glacier area). Despite the different conditions during image acquisitions (e.g., cloudiness, solar altitude and azimuth), the orthophotos and DSMs are consistent and do not show any abrupt colour or brightness changes along the transition of images from different surveys.

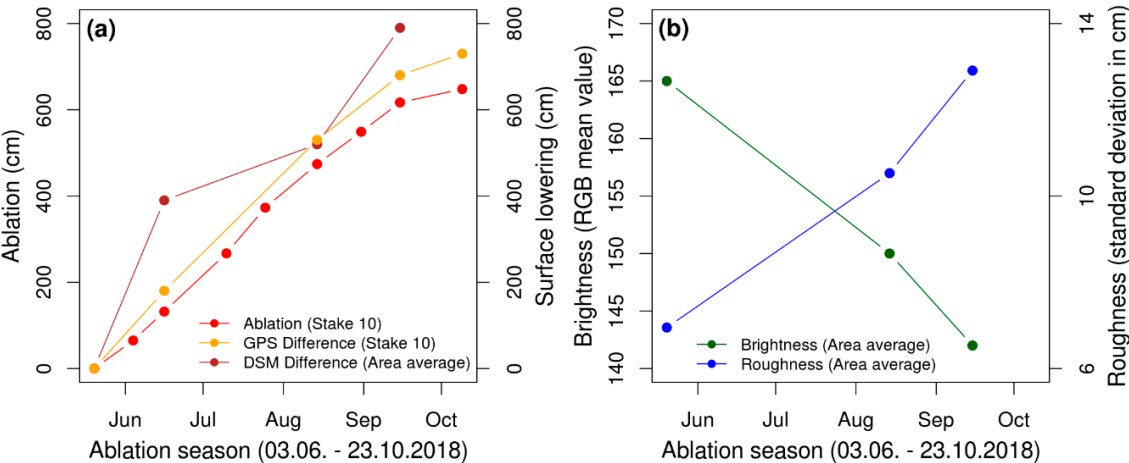

**Figure 10.** Glacier surface changes over the melt season 2018. (**a**) Ablation and GPS surface elevation changes measured directly at stake no. 10 (see Figure 1). Mean surface elevation changes within the comparative area (see dashed rectangle in Figure 9b) derived from OpenDroneMap DSMs. (**b**) Surface brightness and roughness changes within the comparative area derived from consecutive OpenDroneMap orthophotos and DSMs. Brightness and roughness data from 30 June were not considered due to differing illumination conditions caused by cloudiness.

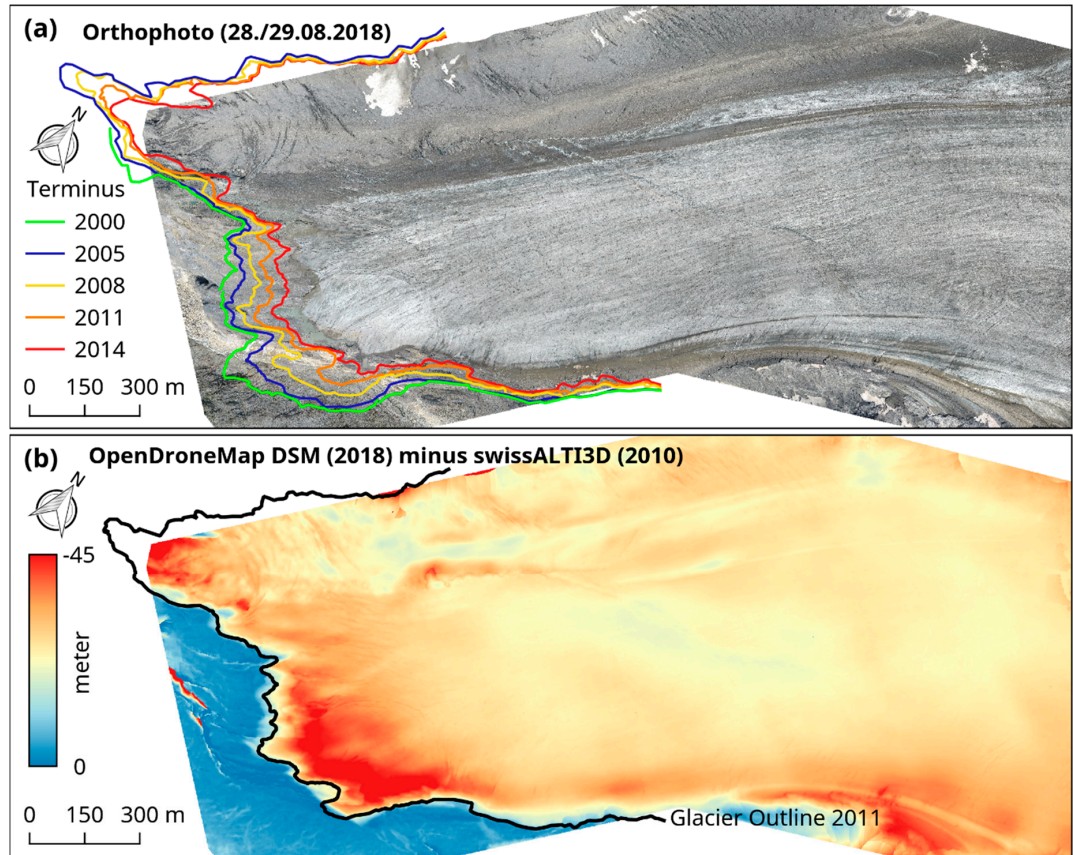

**Figure 11.** (**a**) High-resolution (5 cm) OpenDroneMap orthophoto of the Kanderfirn from 28–29 August 2018 based on 850 aerial images. Five orthomosaics of the Swiss Federal Office of Topography from 2000, 2005, 2008, 2011 and 2014 served for mapping changes of the terminus. (**b**) Glacier surface elevation change (in meters) between 2010 and 2018 based on the difference of two DSMs (OpenDroneMap DSM 2018 minus swissALTI3D 2010).

The June and August orthophotos not only support the analysis and mapping of moraines, crevasses, debris cover, ice cliffs, avalanche cones, dirt cones, meltwater channels, etc., but also accurately allow the determination of the glacier's extent. Over the melt season of 2018, downwasting along the glacier terminus was heterogeneous. The central part of the terminus was relatively stable, whereas the southern part of the tongue "retreated" by more than 40 m, leading to the formation of a small proglacial lake (Figure 8). A similar pattern became apparent when comparing the terminal positions over the last 18 years: the retreat of the glacier terminus between 2000 and 2018 varied from 50 m at the centre to more than 300 m along the northern and southern margins (Figures 11a and A1).

The generated DSM from August 2018 provides recent high-resolution information of the Kanderfirn's surface and can be compared with older DSMs like the swissALTI3D from 2010 to study varying surface elevation changes of the glacier tongue (Figure 11b). Between 2010 and 2018, the glacier surface lowered on average by 25 m, but in general, showed a very variable melt pattern. The largest downwasting of about 45 m occurred at the margin of the debris-free glacier tongue, which translates into an annual vertical ice loss of 5.6 m. In contrast, the downwasting below the Blüemlisalp south face and along the east–west-oriented supraglacial meltwater stream was less pronounced and corresponds to a vertical ice loss of about 20 m in total and 2.5 m per year.

## 5. Discussion

The applied low-cost UAV proved to be appropriate for the acquisition of aerial images and the tested open-source photogrammetry software suitable for the generation of high-resolution orthophotos and DSMs. The quality and accuracy of the consecutive orthophotos and DSMs was sufficient to monitor and investigate glacier surface changes like the increase in surface roughness, decrease in surface brightness and surface lowering over the ablation period. However, compared to the proprietary software tested in this study, the orthophotos and DSMs produced with OpenDroneMap are less accurate.

Small UAVs equipped with open hardware and software of the Paparazzi project have been successfully operated in harsh environments before (e.g., [25]), but here, we demonstrated for the first time that low-cost fixed-wing UAVs are also a reliable remote sensing platform in complex terrain, where topography, local winds, low air pressure and reduced GPS accuracy challenge autonomous flying (e.g., [14]). Other UAV-studies that have so far been performed in glacierized mountain ranges like the Alps and the Himalaya relied on off-the-shelf products and used established proprietary software for the processing of aerial images (e.g., [16–21]). To analyse the dynamics of individual glaciers, these studies compared orthophotos and DSMs from the beginning and the end of the melt period or from consecutive years. Since our study built upon data of multiple UAV surveys, it also allowed for the investigation of intra-seasonal variations like the continuous decrease in surface brightness or increase in surface roughness over the melt period. However, the tested open-source software has not yet reached the standard of established proprietary software in terms of accuracy. The horizontal (XY) accuracy of the OpenDroneMap outputs ranged from 0.7 m to 1.2 m (RMSE) and the vertical accuracy (Z) from 0.7 to 2.1 m, whereas the XY accuracy of Pix4D varied from 0.3 to 0.5 m and the Z accuracy from 0.4 to 0.5 m (RMSE). Furthermore, warping (radial distortion towards the edges) of the OpenDroneMap DSMs was an issue (Figure 6) and hampers the quantification of seasonal or annual surface elevation changes in the range of a few decimetres to meters.

Compared to the application of commercial UAVs and proprietary photogrammetry software, our approach has some limitations, but also several advantages. The conceptualisation and construction of UAVs takes time and assumes familiarization with the topic. Furthermore, operating a UAV requires training and expertise. However, numerous online tutorials and reports facilitate the construction and operation of UAVs using hardware and software of open-source projects like Paparazzi (e.g., [58]). The benefit of self-built UAVs is that they support the implementation of a variety of optical and meteorological sensors [58], are easy to repair and affordable with a limited budget. Regarding the OpenDroneMap software, the major shortcomings based on our experience are the inaccuracies related

to the implemented georeferencing method. The applied affine transformation uses only three out of all available GCPs to align the 3D point cloud and triangular mesh. Increasing the number of GCPs therefore does not necessarily improve the accuracy of the output. Affine transformations include translation, scaling, reflection and rotation, but cannot correct for warping (often referred to as "doming" and "fishbowling"), an effect which can be caused by inaccurate correction of radial lens distortion [59]. Calibrating the deployed camera before starting with the photogrammetric processing is therefore essential to produce highly accurate undistorted images [60]. To achieve a more accurate georeferencing at all available GCPs, the OpenDroneMap developers discussed the implementation of a more advanced approach (e.g. higher-degree polynomials, thin plate spline) [61]. The advantage of OpenDroneMap over proprietary software is its free availability, complete transparency of the source code and fast implementation of new tools and algorithms.

The low-cost approach presented here fosters a variety of glaciological remote sensing applications. Examples are the calculation of surface velocities [21], mapping of surface temperatures [22], derivation of albedo [49] and the detection of small-scale features such as supraglacial meltwater ponds and ice cliffs [20]. In addition to the analysis of individual features and processes, repeated UAV-surveys enable the continuous monitoring of selected glaciers. An arising opportunity for studying the response of benchmark glaciers to global climate change—especially in data-scarce regions—is the comparison of recent UAV-based DSMs with older satellite-based DSMs (see for example Figure 11). For smaller glaciers (e.g., <5 km$^2$), annual geodetic mass balances could be determined by subtracting UAV-based DSMs from beginning and end of the melt period. In this case, it would be necessary to survey the accumulation area as well. UAV-based studies dealing with the quantification of accumulated (winter) snowfall on alpine glaciers are lacking. However, recent studies have demonstrated the potential of snow depth mapping with UAVs in alpine terrain (e.g., [62]) and therefore corroborate the general feasibility of UAV-based snow accumulation and glacier mass surface balance studies. All the aforementioned UAV applications have in common that ground control measurements are necessary to achieve the highest possible accuracy for the obtained products. A density of more than 12 GCPs per km$^2$ is recommended to reduce the horizontal and vertical error to a minimum [57]. Furthermore, the size of the GCP markers should be large enough (≥A2 paper size) for rediscovery on the aerial images. An emerging alternative for laying GCPs is a geolocation approach called GNSS-supported aerial triangulation. The approach uses a high-precision GNSS receiver as the base station to correct the GPS signals recorded by the rover (UAV), either in real time through a wireless connection between base station and rover (RTK = real-time kinematic) or through post-processing after the survey (PPK = post-processed kinematic) [63].

## 6. Conclusions

Our case study from the Kanderfirn in the Swiss Alps demonstrates that self-built fixed-wing UAVs in tandem with open-source photogrammetry software are a powerful low-cost tool to obtain remotely sensed geodata in high spatial and temporal resolution, facilitating the monitoring and investigation of dynamic processes of the earth's surface. The presented method is not limited to glaciological applications and alpine environments. It can be easily transferred to other regions and geoscientific disciplines.

**Data:** The OpenDroneMap orthophotos and DSMs of the Kanderfirn can be downloaded from the open-access repository Zenodo (https://www.doi.org/10.5281/zenodo.2706019). Aerial images from the different surveys are available upon request by email to the first author.

**Supplementary Materials:** The following three tables are available online at http://www.mdpi.com/2076-3263/9/8/356/s1, Table S1: XYZ offset of each orthophoto and DSM at every ground control point, Table S2: Ablation and surface lowering measured over the melt season 2018 at stakes no. 00, 10, 11, 12, 20, 21 and 22, Table S3: XY difference between the UAV-based orthophotos and the SwissImage from 2014, measured at nine reference objects in the glacier forefield.

**Author Contributions:** A.R.G. designed the study. A.R.G. and A.P. developed the fixed-wing UAV. A.R.G., T.J.B., L.M. and S.E. conducted fieldwork and carried out the UAV surveys. C.M.K., T.J.B. and A.R.G. processed the orthophotos and DSMs in OpenDroneMap. All authors were involved in the data analysis. A.R.G. prepared the figures and manuscript with contributions from all co-authors.

**Funding:** This research received no external funding. The expenses for the fieldwork on the Kanderfirn in 2017 and 2018 were covered by the Institute of Geography of the University of Bern.

**Acknowledgments:** We would like to thank Heinz Veit for supporting this study, Manuel Bart for setting up WebODM and Christoph Mayer for help with the photogrammetric processing in Pix4D. Many thanks also go to the supportive Paparazzi UAV and OpenDroneMap communities as well as to the developers of additional open-source software used in this study (e.g. QGIS, R, LibreOffice). We are also grateful to Simon and Daniela Oberli (www.swissglaciers.org) for additional photos of the Kanderfirn and Erika and Toni Brunner of the Mutthornhütte for their hospitality during fieldwork.

**Conflicts of Interest:** The authors declare no conflict of interest.

## Appendix A

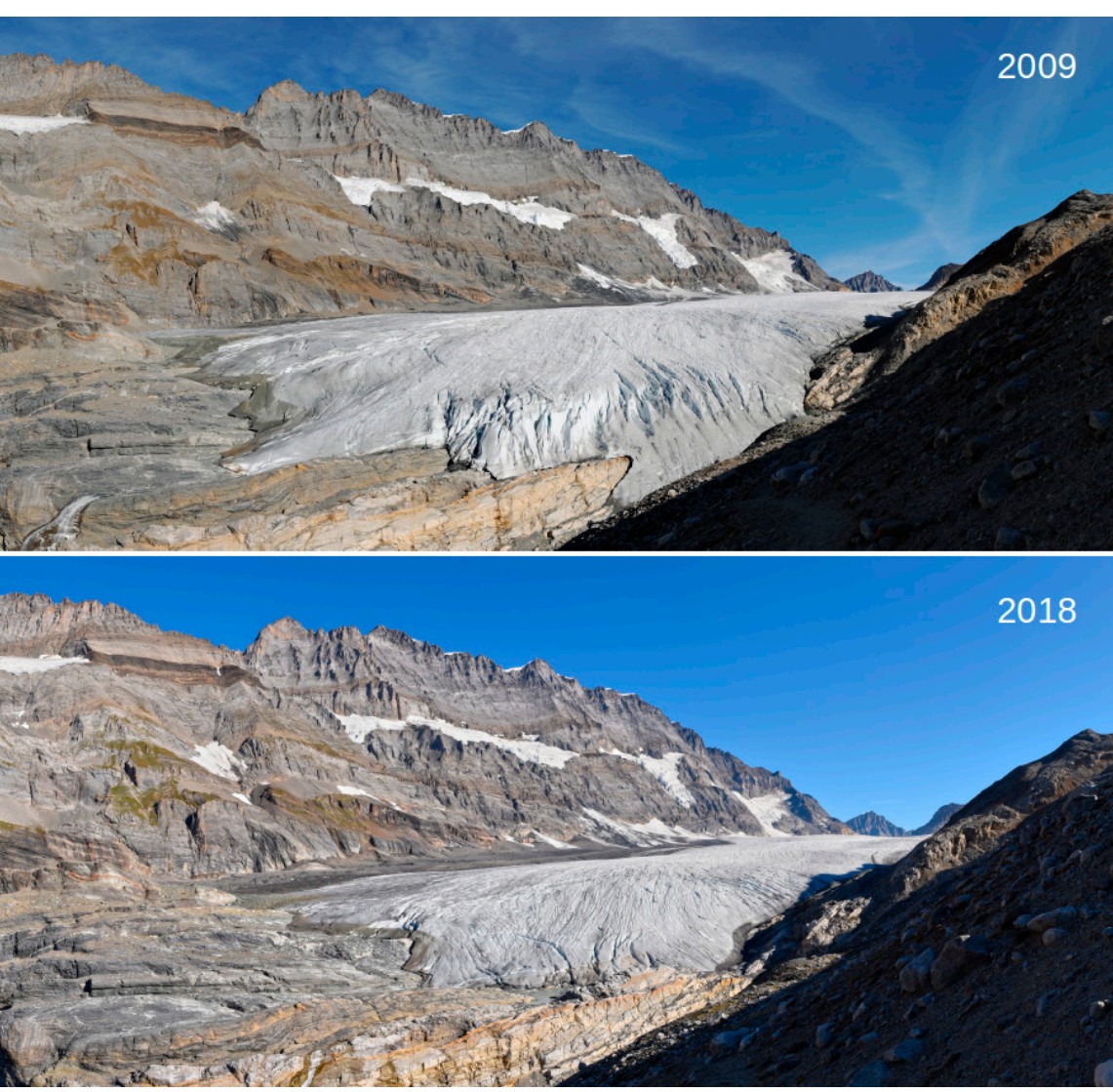

**Figure A1.** Downwasting of the Kanderfirn between 2009 and 2018 (photo credits: Simon Oberli). More comparative photographs of the Kanderfirn are available online (www.swissglaciers.org).

**Table A1.** Individual components of the UAS (the stated costs are only rough estimates).

| UAS Component | Manufacturer | Product | Cost (€) |
|---|---|---|---|
| Flying Wing | EPP-Versand | Knurrus Maximus FPV (140 cm) | 65 |
| Autopilot | Paparazzi UAV | Apogee v.1.0 | 150 |
| Remote Control | Graupner | HoTT mx-16, 2.4 GHz, incl. Receiver | 285 |
| Telemetry Modems | SparkFun | 2 x Xbee Pro S2B 2.4 Ghz, incl. Antenna | 130 |
| Motor | NTM Prop Drive | Series 35-42A 1250Kv 600W | 30 |
| Speed Controler | Turnigy | Plush 60A Speed Controller | 50 |
| Propellers | Aero-Naut | 2 x CAM-Carb. $12 \times 6''$ Folding Propeller | 10 |
| Servo Motors | Multiplex | 2 x Hitec Digital Servos (HS-5245MG) | 70 |
| GPS | Navilock | GPS | 30 |
| Battery | SLS | XTRON 5000mAh 3S1P 11.1V 20C/40C | 50 |
| Camera | GoPro | Hero 5 Black | 430 |

**Table A2.** XY offset between the UAV-based orthophotos and the SwissImage from 2014 measured at nine objects in the glacier forefield (Figure 9a, Table S3). Z offset between the UAV-based DSMs and swissALTI3D from 2010 within a reference area outside the glacier (Figure 7b).

| Date | Software | Version | XY RMSE (m) | Z RMSE (m) |
|---|---|---|---|---|
| 27 September 2017 | ODM | 0.4.1 | 2.5 | 2.0 |
| 3 June 2018 | ODM | 0.4.1 | 0.8 | 1.5 |
| 3 June 2018 | Pix4D | 4.3.31 | 0.9 | 1.8 |
| 30 June 2018 | ODM | 0.4.1 | 0.6 | 0.8 |
| 28/29 August 2018 | ODM | 0.4.1 | 1.2 | 3.4 |
| 29 September 2018 | ODM | 0.4.1 | 0.9 | 0.6 |
| 29 September 2018 | Pix4D | 4.3.31 | 0.7 | 3.3 |

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
