# Peer review of "The Potential of Low-Cost UAVs and Open-Source Photogrammetry Software for High-Resolution Monitoring of Alpine Glaciers: A Case Study from the Kanderfirn (Swiss Alps)"

_geosciences, doi:10.3390/geosciences9080356_

Round 1

Reviewer 1 Report

Review of “The potential of low-cost UAVs and open-source photogrammetry software for high-resolution monitoring of alpine glaciers: A case study from Kanderfirn (Swiss Alps)”

Alexander R. Groos et al.

Overview and key points

Given the significant new and evolving tool of unmanned aerial vehicles in the field of geosciences, the authors test use of low cost UAVs and open source software to analyze the Kanderfirn in the Swiss Alps over two seasons and multiple surveys. Their findings are that low cost UAVs and no-cost software is a viable option for monitoring glaciers . At present the science that they conduct with both methods is not highlighted in abstract/intro, and there are a few places where presenting a more comprehensive analysis would be helpful.

Tools used are:

Paparazzi UAV

OpenDroneMap software

Not mentioned in abstract is that they compared results from ODM with Pix4D (proprietary software). – what were the pros and cons?

The paper is clearly written in general. There are a few places where I would change the order or fix a couple of typos/spelling errors. The authors declare that they do not have conflicts of interest (say with drone or software companies).

My two main recommendations are (1)  that the authors should better quantify the differences between the software examples so that they can clearly state if they are equivalent or if the open software is less good but adequate. They do say that it is adequate but beat around the bush about the detailed comparison. The crux of this is in figure 10 where the software outputs are quite different. A histogram of the differences might show that although it appears significant it is with XX cm overall leading to the assessment of adequate.

(2) Authors should include some of the glaciological findings in the abstract and add the quantifications suggested above into abstract so that it is stronger and more informative.

My overall recommendation is to publish the paper if authors address these two recommendations.

Major comments

The abstract does not indicate any of the scientific results or potential of the results , just that low cost drones are viable. Paper would be stronger if it had some of the science as well as the major finding of the potential of these tools.

229-239 This text suggests that the method of using a cheap/lightweight camera may not be sufficient if your goal is to get calibrated albedo. however if it is to generate a DSM with SfM then it may be adquate. This limitation should be explicit here or in discussion.

Lines 254 specify the dates of the DSMs compared

Line 292 I am surprised that the authors are testing the accuracy of the ODM software at only one site. There are multiple ablation stakes, so there should be the potential to do this in multiple/all places for higher variability (or robustness) of the final results. This would also generate a more statistically valid assessment of the relative accuracy of the programs. The paper will be stronger and more citable if these assessments are included.

Lines 303-306 this is a helpful discussion. May want to add caveat that distance restrictions may vary for other locations/jurisdictions or weather conditions

Figure 6 include a location map showing where these are from? also consider showing a before/after comparison for the changes. it's not obvious if any of these are the same area (I don't think so)

Line 347 (figure 7c) could show all of the different moving window sizes (or a range of small, med, large) as it was not clear in text whether this was a bit arbitrary

Consider swapping figures 9 and 10 (and text about them) – 10 seems to be an overall discussion of the software and the results are quite different. It would be good to identify whether one is better before making results.

Lines 432-436 is it better, good enough, best? the prior figure suggests that it would be good to quantify which one is doing the more accurate job since the DSM differences were quite different. Maybe compare the difference between the software and take out the time issue? then it would highlight whether it was slopes, edges, etc with differences. this is a key factor for saying what approach to use for consistency and accuracy

Line 439 I am more convinced that they offer great images than the robustness of the values. see previous comment

Line 448 quantify the difference between software in line 448

Reviewer 2 Report

The article is well written and complete. just a few remarks:

1) insert also in the abstract the main conclusions in terms of precision of survey achieved

2) Enter the characteristics of the GoPro camera in terms of sensor size and resolution and lens focal length.

3) Better discuss the georeferencing issues of the DSM

4) finally, in the discussion, insert the possibility to correct the calibration errors of the camera into the images before use there in the photogrammetric software
